# Lead Levels in Non-Occupationally Exposed Women with Preeclampsia

**DOI:** 10.3390/molecules26103051

**Published:** 2021-05-20

**Authors:** Katarzyna Gajewska, Marzena Laskowska, Agostinho Almeida, Edgar Pinto, Katarzyna Skórzyńska-Dziduszko, Anna Błażewicz

**Affiliations:** 1Department of Analytical Chemistry, Medical University of Lublin, Chodźki 4a Street, 20-093 Lublin, Poland; annablazewicz@umlub.pl; 2Department of Obstetrics and Perinatology, Medical University of Lublin, Jaczewskiego 8 Street, 20-090 Lublin, Poland; marzenalaskowska@umlub.pl; 3LAQV/REQUIMTE, Laboratory of Applied Chemistry, Department of Chemical Sciences, Faculty of Pharmacy, University of Porto, 4050-313 Porto, Portugal; aalmeida@ff.up.pt (A.A.); ecp@ess.ipp.pt (E.P.); 4Department of Environmental Health, School of Health, P. Porto, R. Dr. António Bernardino de Almeida 400, 4200-072 Porto, Portugal; 5Department of Human Physiology, Medical University of Lublin, Radziwiłłowska 11 Street, 20-080 Lublin, Poland; katarzyna.skorzynska-dziduszko@umlub.pl

**Keywords:** lead, preeclampsia, pregnancy, blood pressure

## Abstract

There are many controversies regarding the relationship between lead exposure andcomplications in pregnancy. Preeclampsia (PE) is a maternal hypertensive disorder which is one of the main causes of maternal and foetal mortality. The aim of our study was to assess blood lead level (BLL) in Polish women with PE (PE group, *n* = 66) compared with healthy, non-pregnant women (CNP group, *n* = 40) and healthy pregnant women (CP group, *n* = 40). BLL was determined by inductively coupled plasma mass spectrometry (ICP-MS). The systolic blood pressure (SBP), diastolic blood pressure (DBP) and BLL in the CP group were significantly lower than in the PE group (*p* < 0.001). Logistic regression analyses of BLL showed a significant positive relationship with the presence of PE. Furthermore, both the SBP and DBP values were positively associated with BLL. This study indicates that preeclamptic women tend to present with significantly higher BLL compared to healthy pregnant women. There were no differences in the BLL between the CP and CNP groups.

## 1. Introduction

Preeclampsia (PE) is a maternal hypertensive disorder, and is one of the main causes of maternal and foetal mortality [1]. The incidence of PE varies greatly worldwide. The global occurrence of PE has been estimated at 4.6%, ranging from 1.0% in the Eastern Mediterranean Region to 5.6% in the African Region [2]. The World Health Organization (WHO) estimates the incidence of PE is seven times higher in developing countries (2.8% of live births) than in developed countries (0.4% of live births) [3]. Other authors report that the prevalence of PE in developing countries ranges from 1.8% to 16.7% [4], where this problem is particularly important, as maternal mortality rates are 20 times higher than those reported in developed countries [5].

The clinical definition of PE has changed over the last 60 years. It is characterised by new onset or worsening hypertension (SBP ≥ 140 mmHg or DBP ≥ 90 mmHg) and proteinuria (greater than or equal to 300 mg/24 h urine collection, or protein/creatinine ratio ≥ 0.3). In the absence of proteinuria, other features indicative of PE are as follows: thrombocytopenia (<100,000 platelets/µL), progressive renal insufficiency (serum creatinine concentration > 1.1 mg/dL or a doubling concentration in the absence of other kidney problems), impaired liver function (abnormally elevated liver enzymes in the blood), pulmonary oedema, and new onset visual or cerebral disturbance [6]. Women suffering from PE are predisposed to convulsions, disseminated intravascular coagulation, placental abruption, cerebral and liver haemorrhage, renal failure, and oedema [7]. The risks to the foetus include hypoxaemia, severe growth retardation, acidosis, premature birth and perinatal death [8,9].

Despite many studies on PE, its aetiology remains uncertain. Early PE (before the 34th week of gestation) is suggested to be the result of decreased blood flow through the placenta, while late PE (after the 34th week of gestation) may be associated with maternal conditions such as hypertension, kidney disease, obesity and diabetes that existed pre-pregnancy [10]. Poor placentation, incomplete uterine remodelling of the spiral arteries during the first few months of pregnancy [11], inability to adapt to inflammatory and cardiovascular changes occurring in pregnancy, immunological intolerance between the fetoplacental unit and maternal tissues, genetic predisposition, nutritional deficiencies [10] and oxidative stress arising from factors released by the placenta [12] are suggested as the main factors associated with the development of PE. The clinical phenotype of PE may be caused by the effects of vascular factors, including excessive production of the soluble form of the vascular endothelial growth factor receptor type 1 (sFlt1) and soluble endoglin (sEng) [10]. Moreover, several risk factors have a potential association with PE: maternal age (younger than 20 or older than 35 years), first pregnancy, multiple gestation, black race, family history of PE or eclampsia, low birth weight of the mother, congenital thrombophilia, low socioeconomic status, diabetes, obesity, genetic defects of the foetus, smoking cigarettes, and many others [13,14].

Oxidant–antioxidant imbalance may have a major role in the pathogenesis of PE, and one of the factors involved in oxidative stress is exposure to toxic metals, e.g., lead [15], which increase the production of free radicals and reduce the availability of bioelements necessary for antioxidant defence mechanisms [16]. Environmental contaminants may interfere with trophoblast cells and lead to poor placentation [13]. Exposure to lead causes oxidative stress and the release of reactive oxygen species (ROS) [17]. Lead also affects the integrity of cell membranes, fatty acid composition and causes the release of lipid peroxidases [18]. Lead exposure causes increased endothelin and thromboxane production, and inhibits ATPase activity and vascular smooth muscle, leading to an increased blood pressure [19]. Excess lead exposure can result in delirium, stupor, coma, hypertension, ataxia, weight loss and peripheral neuropathy, among many other symptoms [20]. Moreover, it can raise the concentration of vasoconstricting prostaglandins and lower vasodilatory prostaglandins. Lead has also been shown to inhibit angiogenesis, reduce endothelial cell growth and cause endothelial injury [21].

Since there are many discrepancies and controversies in the literature regarding the relationship between the lead content of the body and the incidence of PE, the aim of our study was to determine the BLL in women who have not been occupationally exposed to lead, and women with pregnancy complications. Although there are Polish studies concerning lead in pregnancy [22], none of them concerned women with PE. Thus, our research strategy was based on samples collected from women classified as healthy and non-pregnant; healthy and pregnant; and preeclamptic pregnant women who had not been occupationally exposed to lead, living in non-industrial regions in south-eastern Poland.

## 2. Materials and Methods

### 2.1. Study Population

This study was conducted as a joint research project involving the Department of Analytical Chemistry (Medical University of Lublin, Lublin, Poland), the Department of Obstetrics and Perinatology (Medical University of Lublin, Lublin, Poland) and the Department of Chemical Sciences (Faculty of Pharmacy, University of Porto, Porto, Portugal). The study was approved by The Ethics Committee of the Medical University of Lublin (No. KE-0254/251/2016). Informed consent was obtained from patients before blood sampling.

All women came from Lublin Voivodeship (a non-industrial region in south-east Poland). Pregnant women were admitted to the Independent Public Clinical Hospital No 4 in Lublin. Samples were collected for 2 years (2018–2020) from patients during their stay in hospital or during routine testing. For the purposes of our research, we created two control groups (healthy pregnant women and healthy non-pregnant women). Inclusion criteria for the control groups were as follows: normotensive, absence of proteinuria, non-smokers, non-occupationally exposed to lead. As the presence of metabolic syndrome before pregnancy is a known risk factor for thromboembolic complications, premature separation of the placenta, gestational diabetes, placental dysfunction in the form of PE and related complications, in our studies, we excluded patients with this diagnosis. To assess BLL as an independent risk factor for PE, we excluded patients with chronic hypertension, morbid obesity (i.e., BMI > 35 kg/m^2^), or other diseases that could affect blood pressure. A diagnosis of PE was based on the definition from the American College of Obstetrics and Gynecologists [6]. Participants in the PE group were non-smokers, and also non-occupationally exposed to lead.

### 2.2. Collection of Blood Samples

Biochemical analyses were carried out by the ALAB Medical Analysis Laboratory, 8 Jaczewskiego Street, Lublin. For lead determination, fasting venous blood samples (5 mL) were drawn from each individual. A whole blood samples were collected in tubes, with Na_2_EDTA as an anticoagulant. The samples were stored in Eppendorf tubes in a freezer at −25 °C until analysis.

### 2.3. Blood Pressure Measurement

SBP and DBP were measured on the upper arm, in a sitting position, by a physician using a standard blood pressure monitor.

### 2.4. Determination of BLL by ICP-MS

BLL was determined by the use of inductively coupled plasma mass spectrometry (ICP-MS, Thermo Fisher Scientific, Waltham, MA, USA). A volume of 500 µL of blood was appropriately diluted to 5 mL with a diluent solution containing 0.1% HNO_3_, 0.01% Triton X 100 (HPLC grade, Sigma Aldrich, St. Louis, MO, USA)., and internal standards at 10 ppb level (Periodic Table Mix 1, Sigma-Aldrich). Different lead standards solutions (50, 100 and 500 ppb) were prepared from a stock solution of 1000 ppm lead for the calibration. For method validation, the certified reference material BCR-636 lyophilized human whole blood (Sigma-Aldrich) was used. All measurements were conducted in triplicate. Lead concentrations were expressed in µg/dL.

### 2.5. Statistical Analyses

Descriptive statistics were produced for the overall sample population, and also stratified by PE status. As the Kolmogorov–Smirnov and Lilliefors tests indicated that the variables were not normally distributed, the nonparametric Kruskal–Wallis ANOVA test was used for the comparison of continuous variables. A logistic regression analysis was used to evaluate the association between BLL and PE, with the model adjusted for the pregnant women’s age, place of residence (urban/rural), gestational age, multiplicity of pregnancy, as well as the number of previous pregnancies. All analyses were two-tailed with a significance level of 0.05. Statistical analyses were performed using TIBCO Software Inc. (2017) Statistica, v 13.0.0.0 (TIBCO, Tulsa, OK, USA).

## 3. Results

### 3.1. Descriptive Statistics for the Overall Sample Population, and Samples Stratified by PE Status

A detailed characterisation of the population studied, divided into the three groups, is presented in Table 1. As the variables were not normally distributed, nonparametric tests were used for the comparison. Means and standard deviations are presented in Table 1, showing the full characterisation for all groups.

### 3.2. The Kruskal–Wallis ANOVA Test Results of the Comparison of Four Groups

#### 3.2.1. Age

The Kruskal–Wallis ANOVA test results are presented in Figure 1a. The women in the CP group were significantly older than those in the PE group (*p* < 0.001) as well as the CNP group (*p* = 0.024).

#### 3.2.2. Gestational Age

The Kruskal–Wallis ANOVA test results are presented in Figure 1b. The gestational age in the CP group was significantly longer than in the PE group (*p* < 0.001).

#### 3.2.3. Systolic and Diastolic Blood Pressure

The Kruskal–Wallis ANOVA test results are presented in Figure 2a. SBP values in the CP group were significantly lower than in the PE group (*p* < 0.001). There were no differences in SBP values between the CP group and the CNP group (*p* = 0.64). DBP values in the CP group were significantly lower than in the PE group (*p* < 0.001). There were no differences in DBP values between the CP group and the CNP group (*p* = 0.52).

#### 3.2.4. BLL

The Kruskal–Wallis ANOVA test results are presented in Figure 2b. BLL in the CP group was significantly lower than in the PE group (*p* < 0.001). There were no differences in BLL between the CP group and the CNP group (*p* = 1.0).

### 3.3. Logistic Regression Analyses of the Relationship between BLL and the Presence of PE

Logistic regression models showed that the presence of PE was associated with increased odds of blood lead level, as well as the values of systolic and diastolic blood pressure. The presence of PE was also associated with decreased odds of age (Table 2).

## 4. Discussion

In our study, women from the CP group were significantly older than those in the PE group (*p* < 0.001) and the CNP group (*p* = 0.024). Logistic regression models showed that the presence of PE was associated with decreased odds of age. Other studies confirm that PE is more common among younger women and during a first pregnancy [23]. Moreover, maternal age (younger than 20 or older than 35 years) is a potential risk factor of PE [13,14]. The gestational age in the CP group was significantly longer than in the PE group (*p* < 0.001). The control groups included women at various stages of pregnancy, including term-pregnancies, while the delivery of women from the PE group usually occurred prematurely, hence the average gestational age in this group is lower. SBP and DBP values in the CP group were significantly lower than in the PE group (*p* < 0.001). Logistic regression analyses revealed a strong correlation of BLL with SBP and DBP in preeclamptic women (Table 2). There were no differences in SBP and DBP values between the CP group and the CNP group (pSBP = 0.64; pDBP = 0.52).

Lead is one of many established risk factors for hypertension and a possible risk factor for mortality from cardiovascular disease [24]. It is known from animal studies that low doses of lead administered regularly over a long period of time may cause the elevation of blood pressure [25]. Existing research on the effects of lead on the cardiovascular system and blood pressure in humans has shown that elevated blood pressure values are generally more common in people occupationally exposed to heavy metals [26]. However, in women who are not occupationally exposed, the results are inconclusive. Elevated BLL can cause cardiovascular disease such as hypertension by promoting oxidative stress, augmenting adrenergic activity, limiting nitric oxide availability, increasing endothelin production, and promoting inflammation. A study by Gambelunghe et al. [27] suggests that a BLL as low as 2–3 μg/dL increases blood pressure and possibly also the risk of hypertension. According to Skerfving and Bergdahl [28], lead causes an increase in blood pressure in pregnancy at a mean BLL of ≤0.5 μmol/L (i.e., ≤10.36 µg/dL). Recently, it has been shown that elevated BLL can be attributed to a risk of more than 400,000 cardiovascular deaths per year [29]. Despite significant progress in perinatal medicine, pregnancies complicated by arterial hypertension remain a challenge for obstetricians. An increased lead concentration in combination with elevated blood pressure may increase the risk of serious complications in both the mother and the foetus. Some studies suggest that a unit change in BLL leads to a 3.9-fold increase in the risk of hypertension during pregnancy [30], while other studies found only a weak association [31] or even no association between SBP and BLL [32]. A study by Chen et al. [33] showed that higher environmental lead exposure increases the risk of pregnancy-induced hypertension. The findings of Yazbeck et al. [34] also confirm the relationship between BLL and blood pressure, and suggest that environmental lead exposure may play an aetiologic role in pregnancy-induced hypertension. Many other studies of the relationship between elevated BLL or lead exposure and increased blood pressure have yielded controversial results. According to Staessen et al. [35], low levels of lead exposure were not consistently associated with increased blood pressure or the risk of hypertension. Only a weak association between SBP and BLL was found in study by Chu et al. [31]. Lead concentration in whole blood is the best biomarker of lead exposure, with the best ability to discriminate between individuals with different mean concentrations [36]. Among all the works included in an extensive review of chemical elements in PE [37], only 4% of papers considered environmental exposure. Obviously, preventing additional lead exposure can help prevent adverse health outcomes in children [38]. It has been pointed out that lead may cause spontaneous miscarriage at a BLL of about 10μg/dL [28]. Lead crosses the placenta by passive diffusion and can be detected in the foetal brain by the end of the first trimester [39]. During pregnancy, the reabsorption of lead through the bones is increased, which leads to foetal exposure. Greater amounts of lead are released from the bones of smokers and women with low calcium intake. Lead can reach the foetus without difficulty, and the placenta does not protect the foetus from lead poisoning [40]. Elevated lead levels have been associated with gestational hypertension, low birth weight, spontaneous abortion, and impaired neurodevelopment. In 2010, the Centers for Disease Control and Prevention (CDC) published the first guidelines for pregnant and lactating women who have been exposed to lead [38]. The CDC recommends that interventions should begin immediately when the BLL in pregnant women exceeds 5 µg/dL. According to the WHO, the BLL should be lower than 10 µg/dL [17,38]. Increasing maternal BLL by 1 μg/dL is associated with a 1.6% increase in likelihood of PE [41]. The Environmental and Occupational Health Sciences Institute recommends following medical management guidelines for pregnant or breastfeeding women with a BLL ≥ 5 µg/dL [42]. Despite the decline in recent decades, BLL in Polish women of child-bearing age still remains a serious concern [22]. A very recent report on lead pollution in Poland highlighted a need to implement a nationwide lead monitoring program, particularly in pregnant women [43]. There are currently no national recommendations or guidelines for maternity services that include guidance on lead risk assessment and ramifications for the management of pregnancy and lactation. Polish national guidelines on environmental pollution are in line with international guidelines, which recommend intervention if the BLL exceeds 5 µg/dL [38,43].

The present study showed a significant relationship between BLL and PE, although the values recorded are not considered dangerous according to the standards provided by the WHO [38]. The BLL in both CP and CNP groups was similar (*p* = 1.0); however, in the PE group it was significantly higher than in the control groups (*p* < 0.001). Our logistic regression analyses of BLL (Table 2) also showed a significant positive relationship with the presence of PE (*p* = 0.01). These results are consistent with the research of Bayat et al. [17], who observed a higher BLL in PE compared with a healthy group of women (*p* = 0.028). The recent systematic review by Poropat et al. [41] suggested that the BLL is significantly and substantially associated with PE (*p* = 0.005). A cross-sectional study by Motawei et al. [15] also showed a significant relationship between BLL and PE. The mean BLL was 37.68 ± 9.17 µg/dL in the PE group (*n* = 115), compared with 14.5 ± 3.18 µg/dL in the comparison group (*n* = 25) (*p* < 0.001). A study by Disha et al. [44] showed a significantly higher BLL in women with PE (3.42 ± 2.18 µg/dL, n = 23), compared to a healthy, normotensive group (2.38 ± 2.43 µg/dL, *n* = 44). Other studies showing higher lead levels in preeclamptic women compared to healthy women can be found in the literature [16,32,45,46,47,48]. A study by Taylor et al. showed a lower BLL in preeclamptic women compared to a control group (3.63 ± 1.22 µg/dL versus 3.67 ± 1.47 µg/dL), although the difference was not significant [49]. However, in Taylor’s studies, samples were collected before mid-term in pregnancy and BLL in early pregnancy tends to be significantly lower than after mid-term. Some studies suggest that calcium supplementation reduces the risk of both maternal hypertension and PE [50]; therefore, it is expected that pregnant women on such a supplement may have a reduced BLL.

The detailed studies of women’s diets, their mode of nutrition, and their vitamin-mineral supplementation may confirm this relationship in the future. The published research indicates that lead may cause renal disease or hypertension, which implies that the exclusion of women with these symptoms may have reduced study variance, and consequently reduced observed associations between BLL and PE. In order to further support our findings, replication in a larger cohort is needed.

## 5. Conclusions

BLL has an independent and significant association with PE, while there were no differences in the BLL between the CP and CNP groups. Furthermore, both the SBP and DBP values were positively associated with BLL. This study indicates that preeclamptic women tend to present with a significantly higher BLL compared to healthy pregnant women.

To the best of our knowledge, the current study is the first presenting the association between BLL and PE in non-occupationally exposed women living in non-industrial environments. The research findings suggest that despite the lack of occupational exposure, lead, which is commonly present in the general environment, may be significant in the development of the described pregnancy complication.

## Figures and Tables

**Figure 1 molecules-26-03051-f001:**
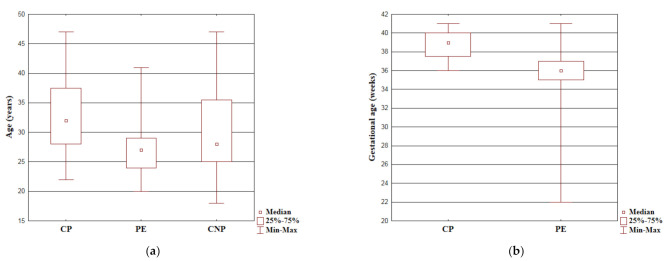
The Kruskal–Wallis ANOVA test results for: (**a**) Age. Years are presented as median values with interquartile ranges (IQR). Min-max are the minimum and maximum recorded values; (**b**) Gestational age. Weeks of gestation are presented as median values with interquartile ranges (IQR). Min-max are the minimum and maximum recorded values.

**Figure 2 molecules-26-03051-f002:**
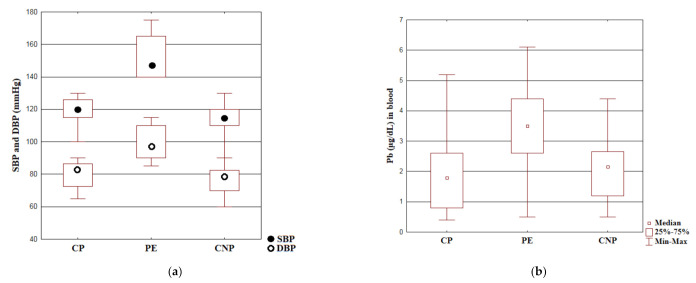
The Kruskal–Wallis ANOVA test results for: (**a**) SBP and DBP values. Values are presented as median values with interquartile ranges (IQR). Min-max are the minimum and maximum recorded values; (**b**) BLL. Concentrations are presented as median values with interquartile ranges (IQR). Min-max are the minimum and maximum recorded values.

**Table 1 molecules-26-03051-t001:** Patient characteristics.

	Mean	Median	Minimum	Maximum	Quartile Range	Standard Deviation
All participants *n* = 146						
Age (years)	29.16	28	18	47	7	5.99
Pb in blood (μg/dL)	2.63	2.6	0.4	6.1	1.8	1.34
SBP (mmHg)	132.7	129.5	90	175	30	20.93
DBP (mmHg)	88	88.5	60	115	15	14.08
Control Non-pregnant (CNP) *n* = 40						
Age (years)	29.5	28	18	47	10.5	7.41
Pb in blood (μg/dL)	2.04	2.15	0.5	4.4	1.45	0.93
SBP (mmHg)	113.9	115.5	90	130	10	8.71
DBP (mmHg)	76.15	79	60	90	12.5	8.83
Control Pregnant (CP) *n* = 40						
Age (years)	32.7	32	22	47	9.5	6.18
Gestational age (weeks)	38.7	39	36	41	2.5	1.49
Pb in blood (μg/dL)	2.04	1.8	0.4	5.2	1.8	1.3
SBP (mmHg)	118.82	120	100	130	11	8.95
DBP (mmHg)	80.5	84	65	90	14	8.12
Preeclampsia (PE) *n* = 66						
Age (years)	26.8	27	20	41	5	3.29
Gestational age (weeks)	35.56	36	22	41	2	2.92
Pb in blood (μg/dL)	3.36	3.49	0.5	6.1	1.79	1.23
SBP (mmHg)	152.62	147.5	140	175	25	12.03
DBP (mmHg)	99.85	97.5	85	115	20	9.62

**Table 2 molecules-26-03051-t002:** Logistic regression analysis of the associations between the presence of PE and BLL. Only statistically significant results are shown.

LOGISTIC REGRESSION			
*n* = 146	^1^ OR	^2^ 95% CI	*p*-Value
*Modeled probability that: PE 1, CP/CNP 0 = 1*			
Blood lead level (µg/dL)	2.65	1.2 to 5.86	0.01
Age (years)	0.7	0.56 to 0.87	0.001
Systolic pressure (mmHg)	2.2	1.25 to 2.93	<0.001
Diastolic pressure (mmHg)	2.0	1.39 to 2.88	<0.001

^1^ OR—odds ratio; ^2^ CI—confidence interval.

## Data Availability

The datasets generated during and/or analysed during the current study are available from the corresponding author on reasonable request; however, some raw data supporting tables are not publicly available in order to protect patient privacy.

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
