# Peer review of "Lead Levels in Non-Occupationally Exposed Women with Preeclampsia"

_molecules, 2021, doi:10.3390/molecules26103051_

Round 1
Reviewer 1 Report
Dear Authors,
This paper is excellent and extremely important. This is the first time in several years, that I don't have any edits to a paper I've been asked to review.
Congratulations on this important work.
Author Response
Thank you very much for the positive commentary about our work.
Reviewer 2 Report
This is an excellent contribution to the literature on Preeclampsia. The framework of the study is crystal clear, the methods sound, and the results easy to follow. The non-parametric statistics are excellent and the figures illustrate the results well. Well done.
Author Response

(The authors gave the same response as above.)

Reviewer 3 Report
The manuscript presents the results of the blood lead level determination in Polish women with preeclampsia compared with healthy, non-pregnant women and healthy, pregnant women. Moreover it describes controversies regarding the role of lead exposure and complications in pregnancy.
In my opinion after a careful survey, only a minor correction are needed:
Abstract: to describe CP group it is enough: healthy, pregnant women (there is no need to add: normotensive). It is also explained in the Materials and Methods, 2.1. Study population.
Table 1. Patient characteristics- in the part describing all participants: gestational age should be excluded, because 40 patients were non-pregnant.
In conclusion, I recommend the work to publish after the corrections indicated above. The manuscript is well written and is suitable for publication in Molecules.
Author Response
Thank you very much for your thorough analysis and all suggestions.
1.Abstract: to describe CP group it is enough: healthy, pregnant women (there is no need to add: normotensive). It is also explained in the Materials and Methods, 2.1. Study population.
Of course, I agree with the opinion, the word 'normotensive' has been deleted
2. Table 1. Patient characteristics- in the part describing all participants: gestational age should be excluded, because 40 patients were non-pregnant.
This is a mistake, the gestational age has been removed as suggested